# Regulatory feedback between VEGF and ERK pathways controls tip-cell expression during sea urchin skeletogenesis

Tovah Nehrer, Tsvia Gildor, Majed Layous and Smadar Ben-Tabou de-Leon*

## ABSTRACT

The sea urchin skeletogenic gene regulatory network (GRN) shows high similarity to the GRN that controls vertebrate vascularization, suggesting that sea urchin biomineralization evolved through co-option of an ancestral tubulogenesis GRN. During vertebrate angiogenesis, vascular endothelial growth factor (VEGF) signaling activates the extracellular-signal regulated kinase (ERK) pathway, which drives gene expression at the tip cells of sprouting blood vessels. Sea urchin VEGF and ERK pathways drive skeletal elongation, but the regulatory interactions between them remain unclear. Here, we reveal positive-feedback circuitry where VEGF signaling activates ERK in the skeletogenic cells near the tips of the skeletal rods, and ERK drives the expression of the VEGF receptor, VEGFR, in these cells. Furthermore, ERK is essential for the transcription of the key skeletogenic transcription factor Ets1/2 and the spicule matrix protein SM50 at the tips of the skeletal rods, while clearing SM50 from the cells at the back. Comparing VEGF and ERK regulation of tip cell expression between vertebrate angiogenesis and sea urchin skeletogenesis illuminates similarities and differences that possibly underlie the co-option of the ancestral tubulogenesis GRN for biomineralization.

KEY WORDS: Co-option, ERK pathway, Biomineralization, Gene regulatory network, Sea urchin, Tubulogenesis, VEGF pathway

## INTRODUCTION

Studies of evolution and development reveal that novel body plans often arise not from *de novo* inventions, but rather from co-option, i.e. the repurposing of pre-existing molecular programs for new functions (Erwin, 2020; True and Carroll, 2002). Co-option is particularly evident in the evolution of biomineralization: the formation of hard mineralized tissue such as shells, bones and teeth (Ben-Tabou de-Leon, 2022; Lowenstam and Weiner, 1989; Murdock, 2020; Weiner and Addadi, 2002, 2011). Biominerals are produced by organisms across all five kingdoms of life that use a variety of different minerals, organic scaffolds, GRNs and biomineralization proteins to form their mineralized structures. The diversity of biominerals and their formation mechanisms have led to the hypothesis that biomineralization evolved independently and

Department of Marine Biology, Leon H. Charney School of Marine Sciences, University of Haifa, Haifa 31905, Israel.

*Author for correspondence (sben-tab@univ.haifa.ac.il)

T.G., 0009-0001-0321-5950; M.L., 0000-0003-2220-0145; S.B.-T.d.-L., 0000-0001-9497-4938

rapidly across different phyla through the co-option of pre-existing developmental programs and the adaptation of ancestral GRNs (Ben-Tabou de-Leon, 2022; Knoll, 2003; Lowenstam, 1981; Lowenstam and Weiner, 1989; Murdock and Donoghue, 2011; Murdock, 2020). Deciphering which elements of the original developmental programs persisted, and which elements changed during the co-option to form mineralized structures is fundamental for understanding the evolution of biomineralization.

A compelling system for studying the regulation and evolution of biomineralization is the sea urchin larval skeleton (Ben-Tabou de-Leon, 2022; Gildor et al., 2021; Morgulis et al., 2019; Oliveri et al., 2008). The larval sea urchin skeleton consists of two interconnected frameworks of calcite rods, known as 'spicules', generated within a tubular cavity produced by specialized skeletogenic cells (Gildor et al., 2021; Oliveri et al., 2008). This tube-shaped structure is obtained when the skeletogenic cells fuse through their filopodia, forming a syncytial network and creating a pseudopodia cable that connects them (Ettensohn and Dey, 2017; Miller et al., 1995). Prior to skeletal formation, the skeletogenic cells arrange themselves into a ring with two lateral cell clusters. Skeletal formation begins within these clusters as mineral-bearing vesicles are deposited into an internal cavity and initially form two triradiate spicules (Miller et al., 1995). The spicules elongate through the deposition of mineral- and matrix-bearing vesicles at the tips of the rods, eventually giving rise to four distinct skeletal rods: body, antero-lateral, post-oral and mid-ventral (Fig. S1; Ettensohn and Malinda, 1993). Therefore, the sea urchin calcite spicules grow via the elongation of a tubular cavity shared by the skeletogenic syncytium and through deposition of mineral ions and matrix proteins at the tips of the spicule rods.

The gene regulatory network (GRN) that controls skeletogenic cell specification and skeletal growth is one of the most elaborate of its kind and shows resemblance to the GRN that drives vertebrate vascularization (Ben-Tabou de-Leon, 2022; Gildor et al., 2021; Morgulis et al., 2019; Oliveri et al., 2008). The transcription factors Ets, Erg, Hex, FoxO and Tel, vascular endothelial growth factor (VEGF), the extracellular signal-regulated kinase (ERK), other signaling pathways, and tubulogenesis genes regulate both sea urchin skeletogenesis and vertebrate vascularization (Adomako-Ankomah and Ettensohn, 2013; Ben-Tabou de-Leon, 2022; Duloquin et al., 2007; Fernandez-Serra et al., 2004; Gildor et al., 2021; Goloe et al., 2024; Morgulis et al., 2019; Oliveri et al., 2008; Röttinger et al., 2004; Shin et al., 2016; Sun and Ettensohn, 2014; Tarsis et al., 2022). The similarity between the sea urchin skeletogenic and vertebrate vascularization GRNs along with their shared tubular structures suggest that the sea urchin skeletogenic GRNs evolved through the co-option of an ancestral tubulogenesis program for biomineralization.

As mentioned, ERK is a key component of both GRNs, playing a crucial role in regulating cell fate specification and structural formation (Chessel et al., 2023; Lavoie et al., 2020; Röttinger et al., 2004; Shin et al., 2016; Zhang et al., 2014). In sea urchin

skeletogenesis, ERK activation occurs in two distinct phases. In the first phase, ERK signaling is activated in skeletogenic precursor cells in a cell-autonomous manner, triggering the expression of Ets transcription factors, which are essential for skeletal cell differentiation. Consequently, when ERK activity is continuously inhibited, no skeleton forms (Chessel et al., 2023; Fernandez-Serra et al., 2004; Röttinger et al., 2004). Following this initial activation, ERK activity is downregulated in the skeletogenic cells until the second phase of ERK signaling during skeletal elongation (Röttinger et al., 2004; Sun and Ettensohn, 2014). The exact timing of this second phase remains unclear; however, previous studies reveal that inhibiting ERK activity after spicule initiation results in shortened skeletal rods and downregulation of some biomineralization genes (Sun and Ettensohn, 2014). Overall, the first phase of ERK activity in the skeletogenic cells is crucial for skeletogenic cell specification; the second phase is crucial for the elongation of the sea urchin skeleton and the expression of biomineralization genes.

In vertebrates, VEGF and ERK regulate endothelial cell specification during vascular development via Ets activation; all three are essential for angiogenesis, i.e. the sprouting of new blood vessels from pre-existing vessels (Chen et al., 2017; Coultas et al., 2005; Gerhardt et al., 2003; Meadows et al., 2011; Srinivasan et al., 2009; Zhang et al., 2014). During angiogenesis, VEGF, ERK and Ets promote endothelial cell proliferation and migration (Chen et al., 2017; Gerhardt et al., 2003; Srinivasan et al., 2009), and VEGF-induced ERK activity is important for the differentiation between the tip cell and stalk cells (Shin et al., 2016; Bridges and Harris, 2016; Gerhardt et al., 2003; Thurston and Kitajewski, 2008). This differentiation process is regulated by ERK-activated Delta-Notch signaling. ERK stimulates the expression of Delta, which leads to tip cell differentiation (Shin et al., 2016). Delta then activates Notch signaling in adjacent cells through cell-junction signaling, which promotes their differentiation into stalk cells (Bridges and Harris, 2016; Gerhardt et al., 2003). In summary, VEGF and ERK regulate endothelial cell behavior and ensure tip-stalk cell patterning during angiogenesis, both of which are crucial for proper blood vessel formation.

VEGF signaling also plays a vital role in sea urchin skeletogenesis with genetic and pharmacological perturbations of this pathway completely abolishing skeletal formation (Adomako-Ankomah and Ettensohn, 2013; Duloquin et al., 2007; Morgulis et al., 2019). During skeletal elongation, VEGF expression is localized to the ectodermal cells near the tips of the post-oral and antero-lateral rods, and guides their elongation. Inhibition of the VEGF receptor (VEGFR) after tri-radiate spicule formation results in shortened skeletons, similar to the effects seen with ERK inhibition at the same stage (Adomako-Ankomah and Ettensohn, 2013; Duloquin et al., 2007; Morgulis et al., 2019; Sun and Ettensohn, 2014; Tarsis et al., 2022). However, whether VEGF activates ERK signaling during sea urchin skeletal elongation remains unclear.

A major difference between vascularization and biomineralization is the mechanical properties of the lumen. In vertebrate vascularization, the lumen is filled with soft, fluid blood while in sea urchin biomineralization the lumen is hard, rigid mineral. A recent study has shown that focal adhesions are formed around the spicules, possibly due to the stiffness of the biomineral (Layous et al., 2025). The activation of focal adhesion kinase (FAK) around the spicules, is mediated through Rho-associated protein kinase (ROCK) activity, and both FAK and ROCK activate ERK activity in skeletogenic cells near the tips of the spicules (Layous et al., 2025). Interestingly, vertebrate FAK, ROCK and ERK are key factors in a mechanosensing circuit that drives the differentiation of vertebrate biomineralizing cells, the osteoblasts, on hard substrates (Kanno et al., 2007; Khatiwala et al., 2009; Salasznyk et al., 2007; Shih et al., 2011).

Thus, sea urchin ERK responds to mechanosensing cues and plays a pivotal role in sea urchin skeletogenesis, yet its interaction with VEGF signaling and the skeletogenic GRN during skeletal elongation require further investigation. Here, we study the timing, regulation and role of ERK signaling during skeletal elongation in the sea urchin species *Paracentrotus lividus* (*P. lividus*). Our findings illuminate the intricate interactions between VEGF and ERK pathways that drive gene expression at the tips of the skeletal rods, and highlight ERK as a possible integrator of multiple spatio-temporal inputs.

## RESULTS

### Inhibition of ERK after spicule formation primarily blocks the elongation of the post-oral rods

The larval sea urchin skeleton consists of four primary skeletal rods: the body, mid-ventral, antero-lateral and post-oral rods. Previous studies have shown that inhibiting ERK activity after spicule formation results in shortened skeletal rods (Röttinger et al., 2004; Sun and Ettensohn, 2014). However, when ERK activity starts to influence skeletal morphology remains unclear. To pinpoint this timing, we inhibited ERK activity by applying the U0126 inhibitor at 25 h post-fertilization (hpf), which is immediately after the formation of triradiate spicules in *P. lividus* (Fig. 1A,G). This inhibitor blocks MAPK/ERK kinase (MEK), the sole activator of ERK, and was previously used to successfully block sea urchin MEK (Fernandez-Serra et al., 2004; Röttinger et al., 2004; Sun and Ettensohn, 2014). After applying this inhibitor, we compared the skeletal morphology of control and treated embryos to determine when ERK-inhibited embryos began to differ from the control group.

At 30 hpf, both control and MEK-inhibited embryos appeared indistinguishable from each other (Fig. 1B,H,M). However, by 32 hpf, control embryos began to show nascent post-oral rod development, while MEK-inhibited embryos lacked this initial rod formation (Fig. 1C,I,N). By 34 hpf, the difference between the two groups became statistically significant (Fig. 1R), with the majority of control embryos having elongated post-oral rods, whereas the majority of MEK-inhibited embryos still exhibited no post-oral rod development (Fig. 1D,J,O). By 36 hpf, all of the skeletal rods in control embryos were elongated and visible, while MEK-inhibited embryos had smaller rods and still nascent to no post-oral rod development (Fig. 1E,K,P). These same phenotypes were also present at 40 hpf, with control embryos showing fully elongated rods, while MEK-inhibited embryos exhibited shortened rods, in particular nascent and underdeveloped post-oral rods (Fig. 1F,L,Q). These observations show that ERK activity is required for proper skeletal elongation, in particular for the timely initiation and elongation of the post-oral rods, with apparent skeletogenic phenotypes from 32 hpf and onwards.

### Activity of ERK in the skeletogenic cells during skeletal elongation

Next, we aimed to investigate the onset and localization of ERK activity within the skeletogenic lineage during normal skeletal elongation. To do this, we examined ERK activity under control conditions using immunostaining for dually phosphorylated ERK (dpERK) and the skeletogenic cell marker 6a9 (Ettensohn and McClay, 1988) at various stages of skeletal development (Fig. 2). ERK activity was observed in multiple embryonic territories, including both skeletogenic and non-skeletogenic cells (Fig. 2A-P).

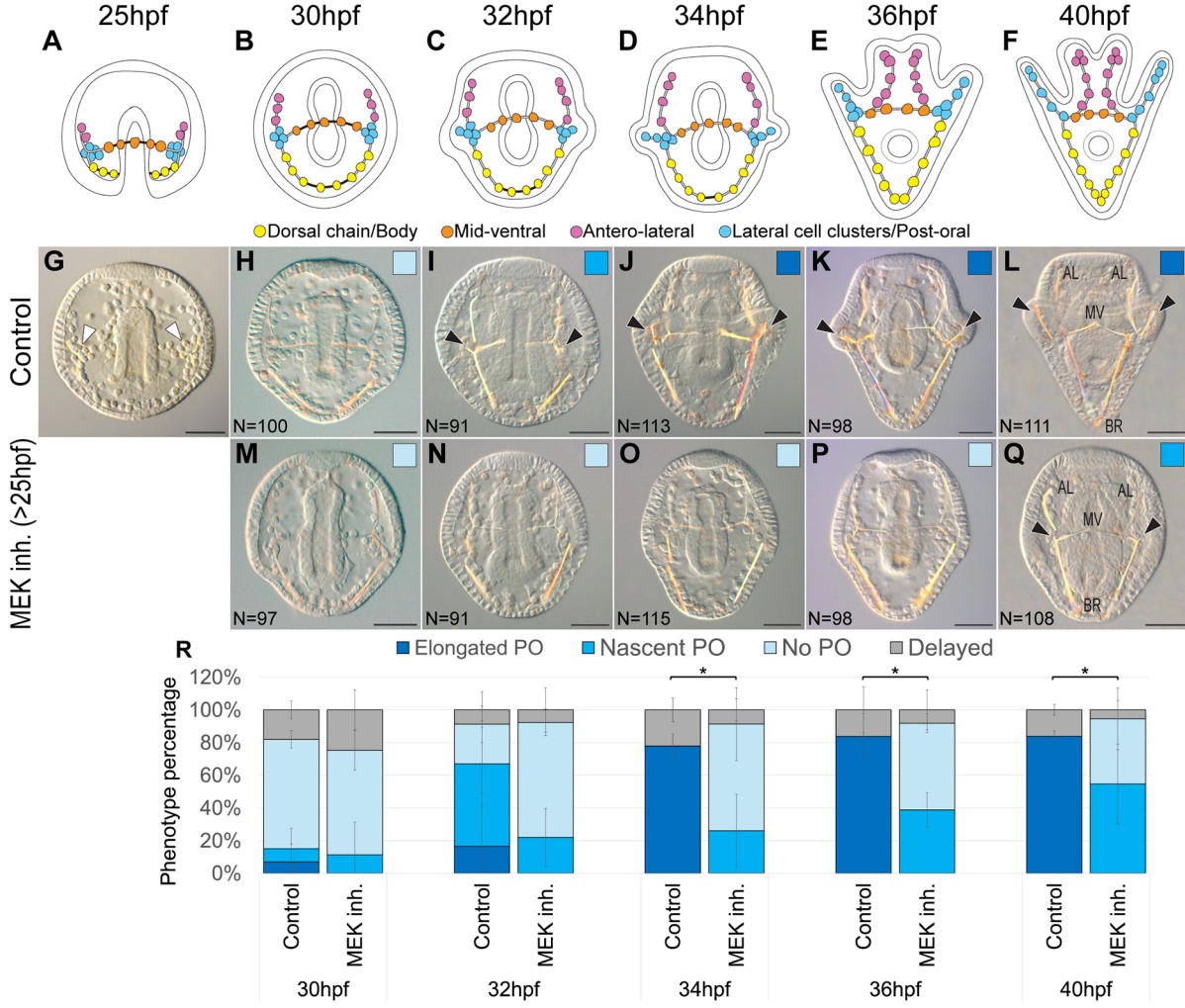

**Fig. 1. Inhibition of ERK activation after spicule formation (>25 hpf) visibly affects skeletal growth from 32 hpf onwards.** (A-F) Illustrations of normal skeletal development, where the skeletogenic cells are color coded according to their spatial localization. Cells of the dorsal chain and the body rods (BR) are marked in yellow, cells in the mid-ventral (MV) rods are marked in orange, cells along the antero-lateral (AL) rods are marked in pink, and cells in the lateral cell clusters and the post-oral (PO) rods are marked in blue. (G-Q) A time-course comparison of skeletal development across six time points, where the dominant skeletogenic phenotype is presented for (G-L) control embryos and (M-Q) embryos where 10 μM of U0126 was added at 25 hpf. The following time points are shown: (A,G) 25 hpf, (B,H,M) 30 hpf, (C,I,N) 32 hpf, (D,J,O) 34 hpf, (E,K,P) 36 hpf and (F,L,Q) 40 hpf. The white arrowheads indicate the triradiate spicules, while the black arrowheads indicate the post oral rods. (R) Graphs quantifying the percentage of each phenotype in control and MEK-inhibited embryos. Color code corresponds to the phenotypes shown in H-Q. Data are mean±s.d. Non-parametric Mann–Whitney U-tests were run to compare the percentage of the prominent phenotype in control embryos to that of the same phenotype in treated embryos. Treatment phenotypes are significantly different compared to control from 34 hpf onwards, with *P=0.037 for 34 hpf, 36 hpf and 40 hpf. Experiments were conducted in three independent biological replicates. Scale bars: 50 μm.

At 25 hpf, when the tri-radiate spicules begin to form in the lateral cell clusters, dpERK activity in the skeletogenic cells was relatively low (Fig. 2A-D,Q). However, by 30 hpf, we observed a marked increase in dpERK activity in the skeletogenic cells along all skeletal structures (Fig. 2E-H,Q). At 36 hpf, the dpERK signal became more localized, particularly to the skeletogenic cells near the tips of the rods (Fig. 2I-L). Notably, dpERK activity was most prominent in the growing post-oral rods, while activity in the body and antero-lateral rods was reduced (Fig. 2I-L,Q). By 40 hpf, ERK activity was again observed in cells near the tips of all skeletal rods (Fig. 2M-P), with an increase in the body and antero-lateral rods, although activity remained most prominent in the post-oral rods (Fig. 2Q). Quantification of dpERK activity at these time points showed heightened ERK activity by 30 hpf and onward, both across the total number of embryos scored and within the specific skeletal rods (Fig. 2Q). Thus, ERK activity in the skeletogenic lineage

increases around 30 hpf, and localizes to the tips of the rods by 36 hpf. This timing, along with the sustained high ERK activity in the post-oral rods, aligns with the observed skeletal phenotypes from 32 hpf onward (Fig. 1).

## VEGF signaling activates ERK in the skeletogenic cells during a skeletal elongation

After determining the timing and effects of the second active phase of ERK, we sought to investigate the regulation of this activity by VEGF signaling. Previous studies have suggested a potential link between these pathways, as inhibiting both VEGF and ERK signaling after spicule initiation results in truncated skeletal rods and similar changes in the spatial expression of skeletogenic genes (Adomako-Ankomah and Ettensohn, 2013; Sun and Ettensohn, 2014). Yet, a regulatory link between VEGF signaling and ERK activity has not been shown before. To further explore the

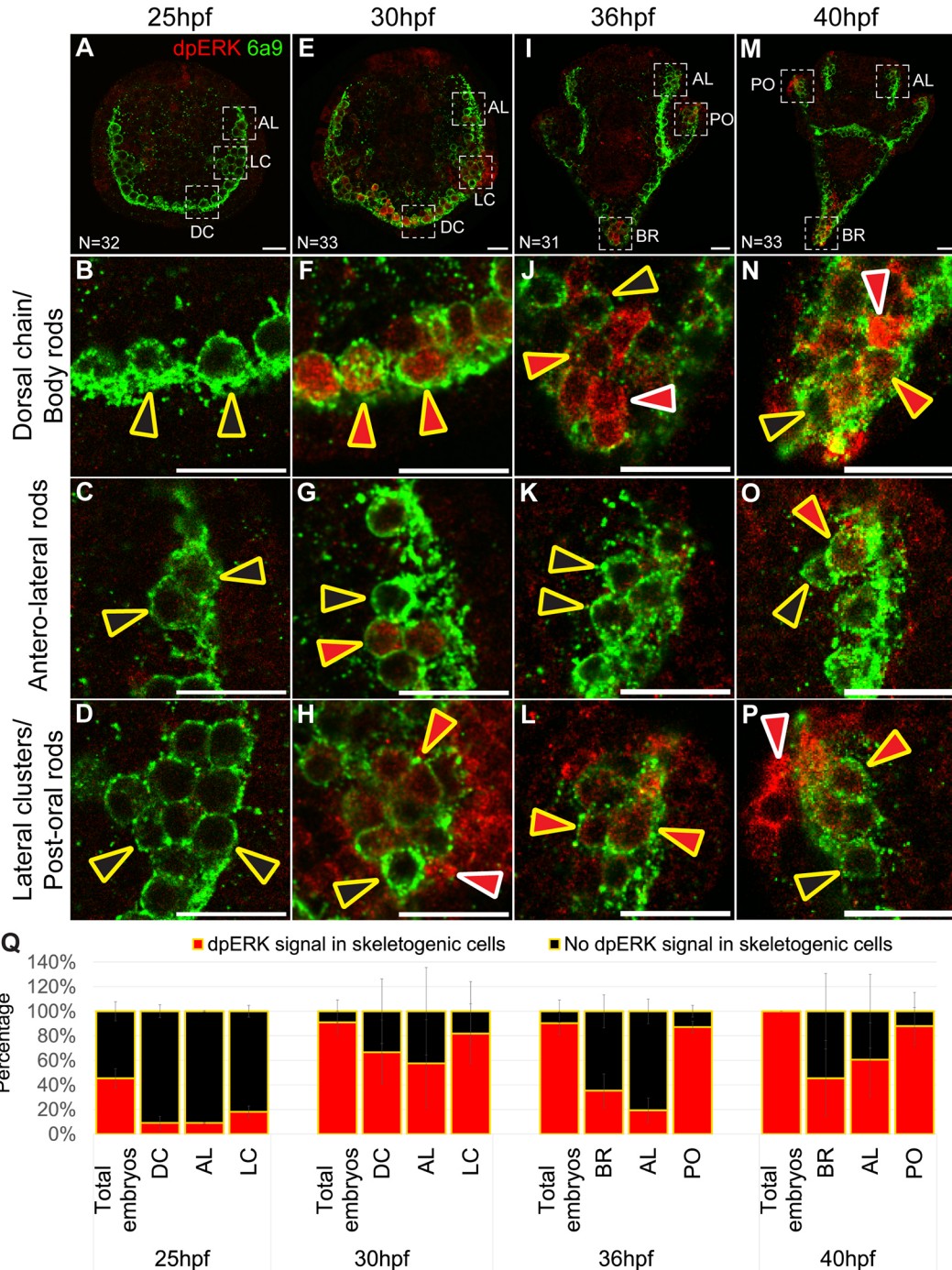

**Fig. 2. ERK activity in the skeletogenic cells is upregulated from 30 hpf and is localized at the tips of the rods, starting from 36 hpf.**
(A-P) Representative images showing immunostaining of sea urchin embryos at different developmental stages with dually phosphorylated ERK (dpERK) in red and the skeletogenic membrane marker 6a9 in green. (A,E,I,M) Whole embryos at the different time points. (B-D,F-H,J-L,N-P) Higher magnifications of specific skeletal structures: dorsal chain/body rods (DC/BR), the antero-lateral rods (AL) and lateral cell clusters/post-oral rods (LC/PO). Red arrowheads outlined in yellow indicate skeletogenic cells with active ERK; red arrowheads with white outlines indicate non-skeletogenic cells with active ERK; black arrowheads outlined in yellow point to skeletogenic cells lacking ERK activity. (A,E,I,M) The numbers in the bottom left corner indicate the total number of embryos assessed at that time point. (A-D) 25 hpf, (E-H) 30 hpf, (I-L) 36 hpf and (M-P) 40 hpf. (Q) Graphs showing the percentage of embryos with a dpERK signal in skeletogenic cells versus those with no dpERK signal in the skeletogenic cells. This was measured across all embryos scored and within the specific skeletal rods. Experiments were conducted in three independent biological replicates. Data are mean±s.d. Scale bars: 20 μm.

relationship between these two pathways, we applied VEGF and ERK inhibitors after skeletal initiation (>25 hpf) and examined ERK activity using immunostaining for dpERK and 6a9, the skeletogenic cell membrane marker (Fig. 3). In our experiments, we used the VEGFR inhibitor Axitinib, which was previously used

in sea urchins and resulted in similar skeletogenic phenotypes to those of VEGFR genetic perturbations (Adomako-Ankomah and Ettensohn, 2013; Morgulis et al., 2019; Sun and Ettensohn, 2014). We observed ERK activity at 40 hpf, when, based on our previous findings, all embryos showed ERK activity in the skeletogenic cells

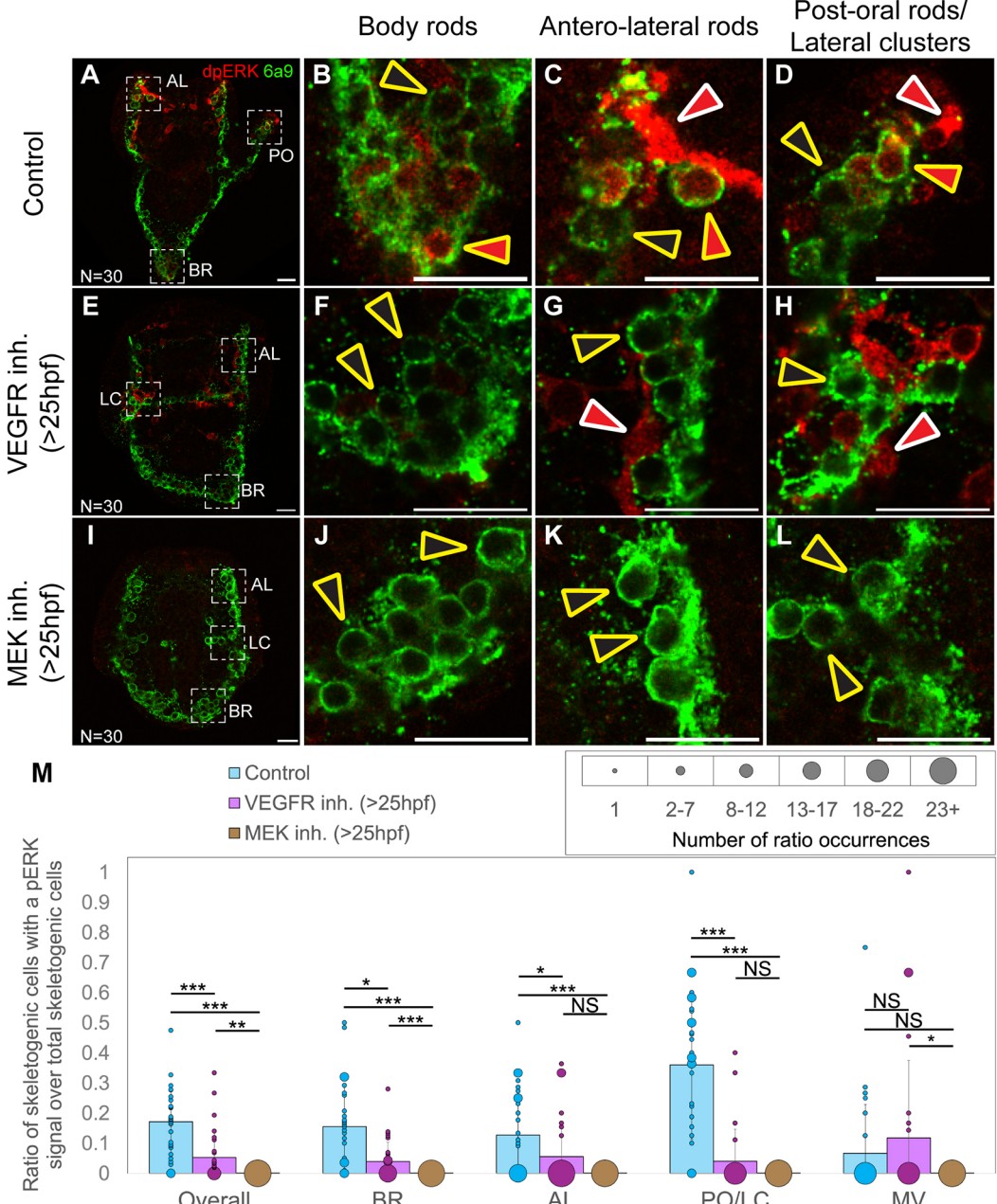

**Fig. 3. VEGF signaling activates ERK specifically in the skeletogenic cells.** Immunostaining of dpERK in red and 6a9 in green for control embryos (A-D), VEGFR-inhibited embryos (150 nM Axitinib >25 hpf, E-H) and MEK-inhibited embryos (10 μM U0126>25 hpf, I-L) at 40 hpf. (A,E,I) Whole embryos; (B-D,F-H,J-L) Higher magnifications of specific skeletal structures: the body rods (BR), the antero-lateral (AL) rods, lateral (cell) clusters/post-oral (LC/PO) rods. Red arrowheads outlined in yellow indicate skeletogenic cells with active ERK; red arrowheads with white outlines indicate non-skeletogenic cells with active ERK; black arrowheads outlined in yellow indicate skeletogenic cells lacking ERK activity. (A,E,I) The numbers at the bottom left of the panels indicate the total number of embryos assessed at that time point. (M) Graphs quantifying the ratio of skeletogenic cells with dpERK signals to the total number of skeletogenic cells seen in each rod and overall. Data are mean ratio±s.d.; circle diameter represents the number of ratio occurrences observed at that value, as indicated by the scale above the graphs. Statistical significance was measured using Kruskal–Wallis tests (non-parametric one-way ANOVA) along with pairwise tests between groups, where NS indicates not significant, *$P<0.05$, **$P<0.01$ and ***$P<0.001$. Experiments were conducted in three independent biological replicates. Scale bars: 20 μm.

(Fig. 2Q). As expected, MEK inhibition leads to a complete loss of ERK activity in all cell types (Fig. 3I-L). In contrast, VEGFR inhibition specifically reduces dpERK levels in the skeletogenic cells, but not in neighboring cells (Fig. 3E-H).

During these experiments, we noted considerable variability in the dpERK signal between individual embryos, and even between homologous rods within the same embryo, in agreement with

previous studies (Layous et al., 2025). This variation could be a result of the highly dynamic activity of ERK, as reported in other systems (Aoki et al., 2017; Gagliardi et al., 2021), yet it requires a quantification of the dpERK signal to assess the effect of the two inhibitors compared to control. To that end, we measured the number of skeletogenic cells with dpERK activity in each rod and divided it by the total number of skeletogenic cells per rod. In

embryos treated with VEGFR and MEK inhibitors, the post-oral rods fail to form. Therefore, under these conditions, instead of counting the cells along the post-oral rods, we counted the cells in the lateral cell clusters, from which these rods would normally form. Our analysis (Fig. 3M) revealed that VEGFR inhibition caused a significant reduction in ERK activity in the body (*P*=0.03), antero-lateral (*P*=0.03) and post-oral (*P*<0.001) rods when compared to control embryos. Similarly, MEK inhibition resulted in a significant reduction in ERK activity in all these rods (all *P*<0.001) compared to controls. However, no significant difference in ERK activity was observed between VEGFR and MEK inhibition in the post-oral and antero-lateral rods, although a significant reduction was found in the body rods (*P*<0.001). This suggests that while VEGFR inhibition affects ERK activity in the body rods, the impact is less pronounced than with MEK inhibition. Further analysis of this quantification revealed that the total number of cells in each rod did not decrease with either inhibitor, except in the post-oral rods (Fig. S2). In conclusion, these results demonstrate that VEGF signaling activates ERK specifically in skeletogenic cells near the tips of the growing rods during skeletal elongation.

### VEGFR and MEK inhibition similarly affect skeletogenic gene expression level

We next aimed to compare the effects of VEGF and ERK signaling on gene expression levels during skeletal elongation. While previous studies have shown that both pathways influence gene expression during sea urchin skeletogenesis (Morgulis et al., 2021; Sun and Ettensohn, 2014; Tarsis et al., 2022), we sought to determine whether they induce similar changes when compared directly. To investigate this, we inhibited VEGFR and MEK activity and measured changes in gene expression using qPCR. Both inhibitors were applied at 25 hpf, and gene expression was analyzed at 36 hpf, a time when ERK skeletogenic phenotypes are clearly apparent (Fig. 1). Our results show that inhibiting both VEGFR and MEK leads to a reduction in the expression of key skeletogenic regulatory genes. This includes genes encoding the transcription factors Ets1/2 and Pitx, as well as the *VEGFR* gene (Fig. 4). *pitx* is expressed at the tips of the post-oral rods

that fail to form when VEGF and ERK are inhibited, which could explain the abrupt reduction in its expression (Sun and Ettensohn, 2014; Tarsis et al., 2022). In addition, the expression levels of two spicule matrix proteins, SM30 and SM50, show minor but significant reduction under MEK inhibition, while VEGFR inhibition only mildly effects SM50 expression. VEGFR inhibition did not affect the expression of the tested non-skeletogenic genes, while MEK inhibition had minor effects on genes expressed in other embryonic regions, such as the dorsal ectoderm (*dlx*), endoderm (*foxa*) and pigment cells (*GCM*). Overall, these findings suggest that VEGF and MEK pathways converge on the regulation of skeletogenic gene expression.

### ERK signaling regulates the spatial expression of regulatory and biomineralization genes

During skeletogenesis, specific genes exhibit differential expression patterns, despite the syncytial network of the skeletogenic cells. Notably, the expression of key regulatory and biomineralization genes becomes localized to the cells near the tips of the growing rods, the regions of active skeletal growth (Chang and Su, 2022; Morgulis et al., 2021; Sun and Ettensohn, 2014; Tarsis et al., 2022). Here, we sought to explore the effect of ERK activity on the dynamic spatial expression of key skeletogenic genes. We inhibited ERK activity at 25 hpf and examined the spatial expression of skeletogenic genes using whole-mount *in situ* hybridization (WMISH) from 28 hpf (2 h before the upregulation of ERK activity in skeletogenic cells; Fig. 2) through 40 hpf. We tested the response of two regulatory genes, *ets1/2* and *VEGFR*, the vertebrate homologs of which regulate vascularization (Chen et al., 2017; Watanabe et al., 2004; Wythe et al., 2013), and two biomineralization genes encoding the spicule matrix proteins SM30 and SM50, which are specific to sea urchins (Kitajima et al., 1996; Wilt et al., 2013).

Our results demonstrate that ERK activity is crucial for the maintenance of *ets1/2* and *VEGFR* activity in the skeletogenic cells (Fig. 5). At 28 hpf, there is no observable effect of ERK inhibition on the expression patterns of *ets1/2* and *VEGFR* in the skeletogenic cells (Fig. 5A,E,I,M). At 32 hpf, *ets1/2* expression in control

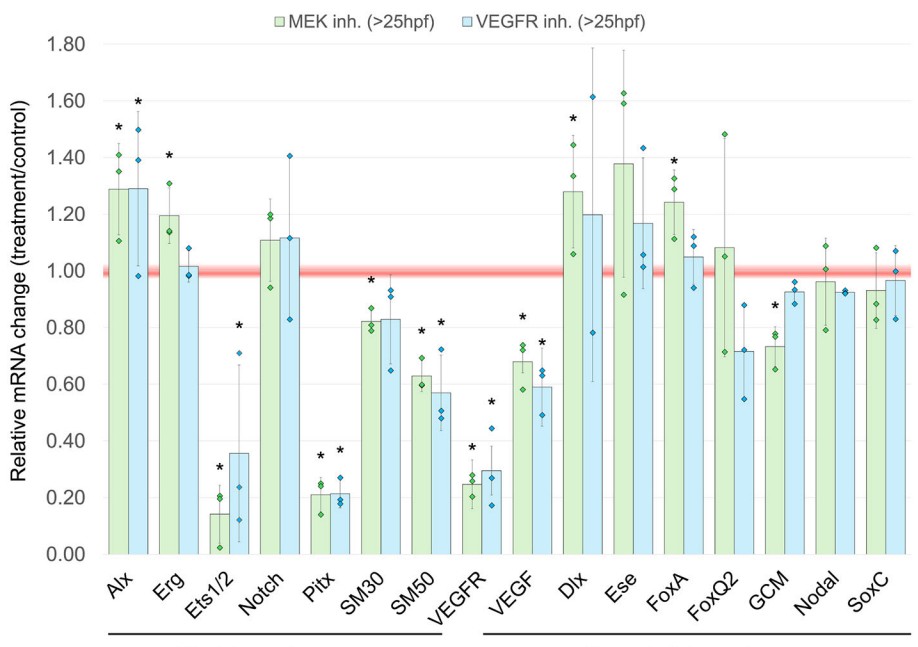

**Fig. 4. Inhibition of VEGF and ERK activities leads to similar changes in skeletogenic gene expression.** Relative gene expression levels in MEK-inhibited embryos (10 µM U0126>25 hpf) and VEGFR-inhibited embryos (150 nM Axitinib >25 hpf) compared to the control DMSO-treated embryos, measured by qPCR at 36 hpf. A ratio of one (indicated by the red line) represents the unaffected gene expression level under treatment, below one indicates downregulation with treatment and above one indicates upregulation with treatment. Data are mean±s.d. Diamonds represent the results of the individual replicates. *P<0.05 (one-tailed z-test). Experiments were performed in three biological replicates, except for *Dlx*, where two replicates were performed.

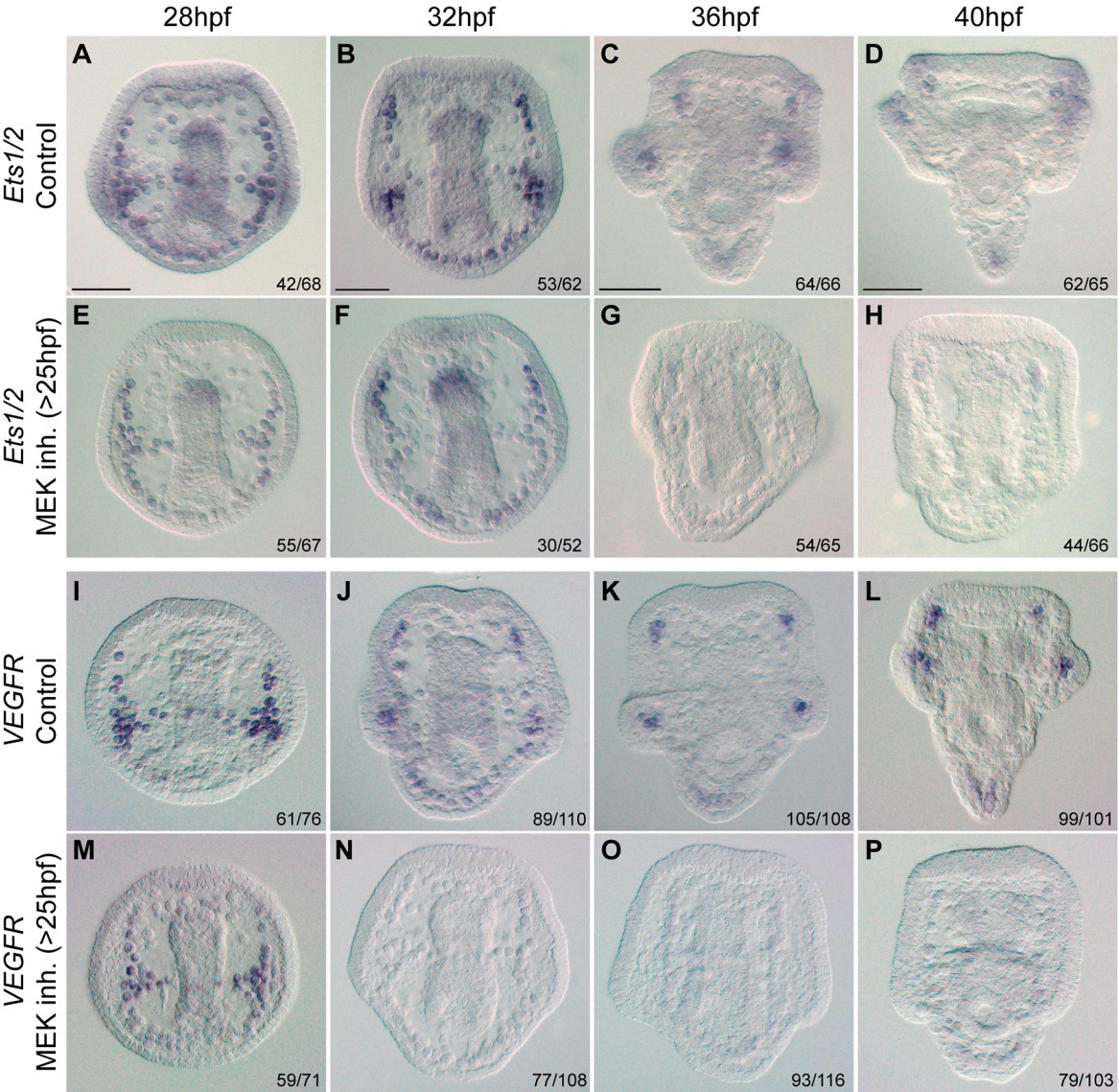

**Fig. 5. ERK signaling is required for normal expression of regulatory skeletogenic genes.** (A-P) Representative images showing the spatial expression of genes *ets1/2* (A-H) and *VEGFR* (I-P) under control (A-D,I-L) and MEK-inhibition (10 μM U0126>25 hpf, E-H,M-P) conditions. Numbers on the bottom right of each panel indicate the number of embryos seen with the spatial expression pattern shown in the image over the sample size. Other embryos showed mild differences from the expression patterns reported here, either stronger or weaker staining in some rods, that can be explained by the asynchronicity of the cultures. A complete dataset with all expression patterns at all time points is provided in Fig. S3. Experiments were conducted in two (*ets1/2*, *VEGFR* at 28 hpf) or three (*VEGFR* at 32, 36 and 40 hpf) independent biological replicates. Scale bars: 50 μm.

embryos is enriched in the lateral cell clusters that begin to form the post-oral rods, in the tips of the antero-lateral rods, as well as along the dorsal chain (the origin of the body rods, Fig. 5B). In contrast, *ets1/2* expression in MEK-inhibited embryos is lost in the lateral cell cluster and is restricted to the dorsal chain and antero-lateral rods (Fig. 5F). At this time, in control embryos, the spatial expression of *VEGFR* is similar to the pattern of *ets1/2* (Fig. 5J), while under MEK inhibition, *VEGFR* expression is almost completely lost (Fig. 5N). By 36 hpf and beyond, control embryos show *ets1/2* and *VEGFR* expression strongly localized to the tips of the antero-lateral and post-oral rods, with weaker signal at the tips of the body rods (Fig. 5C,D,K,L). However, under MEK-inhibition, the expression of both *ets1/2* and *VEGFR* is almost abolished (Fig. 5G,H,O,P). Thus, *ets1/2* and *VEGFR* expression at cells near the skeletal growth zones strongly depends on ERK activity during skeletal elongation.

In contrast, MEK inhibition did not affect the expression of *SM30* but did alter the spatial distribution of *SM50*. In control embryos, *SM30* is expressed across all skeletogenic cells, except for those along the mid-ventral rods, at all timepoints (Fig. 6A-D). This pattern remained largely unchanged under MEK inhibition, except that *SM30* expression was absent from the post-oral rods, as MEK inhibition blocks the elongation of these rods (Fig. 6E-H). This suggests that *SM30* regulation is independent of ERK signaling.

*SM50* expression is enriched in the skeletogenic cell clusters and the antero-lateral rods at 28 hpf, in both control and MEK inhibited embryos (Fig. 6I-M). By 32 hpf, *SM50* expression in control embryos is strongly upregulated at the cells near the tips of the antero-lateral rods, in the lateral cell clusters, and along the dorsal chain and mid-ventral rods (Fig. 6J). However, in MEK-inhibited embryos, these spatial expression patterns are lost, and *SM50* expression became uniformly expressed across the entire

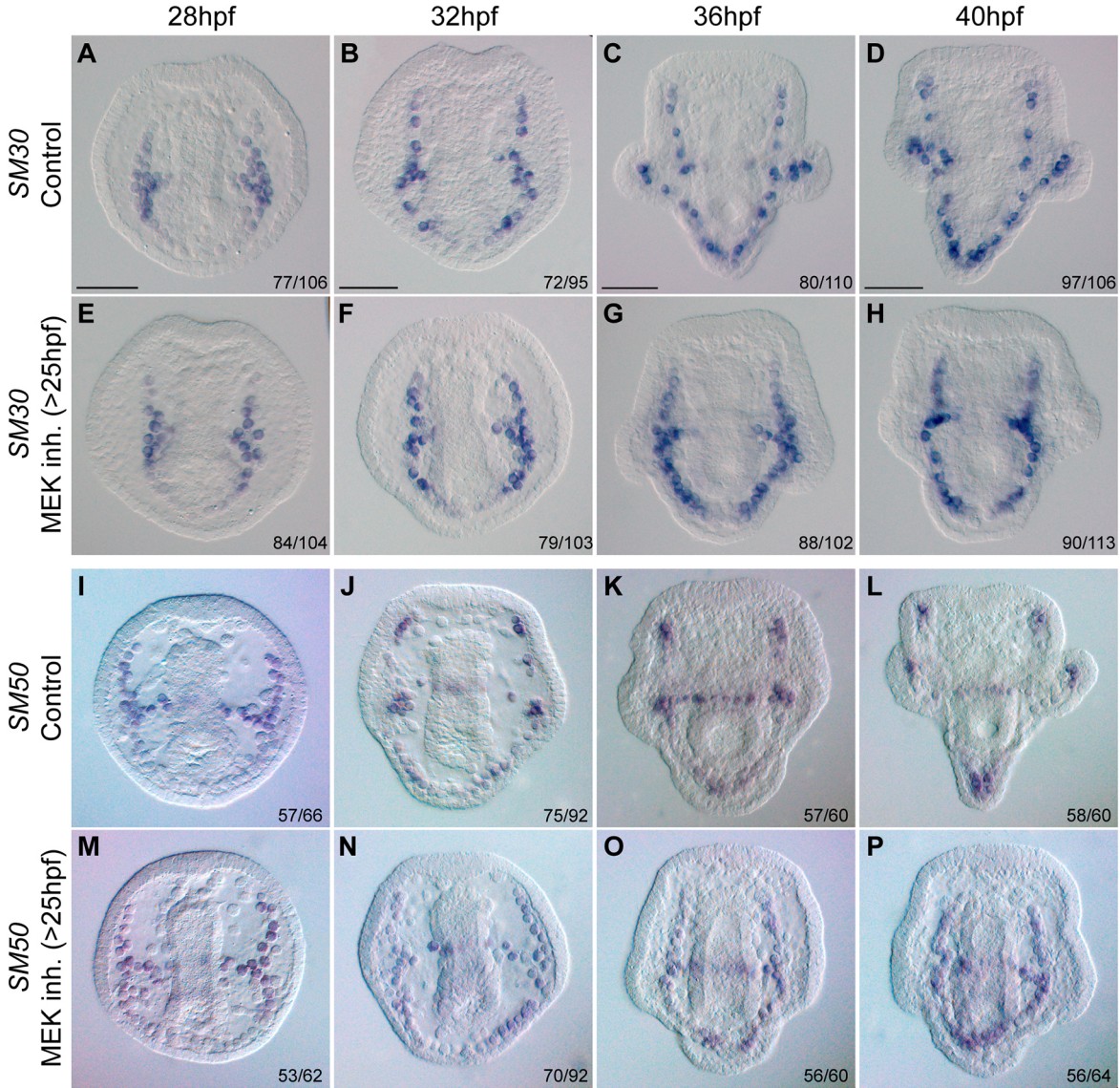

**Fig. 6. ERK signaling is required for normal spatial expression of SM50 but not SM30.** (A-P) Representative images showing the spatial expression of *SM30* (A-H) and *SM50* (I-P) genes in control (A-D,I-L) and MEK-inhibition (10 μM U0126>25 hpf, E-H,M-P) conditions. Numbers in the bottom right of each panel indicate the number of embryos seen with the spatial expression pattern shown in the image over the sample size. Other embryos showed mild differences from the expression patterns reported here, either stronger or weaker staining in some rods that can be explained by the asynchronicity of the cultures. A complete dataset with all expression patterns at all time points is provided in Fig. S4. Experiments were conducted in two (*SM50* at 28, 36, 40 hpf) or three (*SM50* at 32 hpf, *SM30*) independent biological replicates. Scale bars: 50 μm.

skeletogenic cell population (Fig. 6N). By 36 hpf and onward, control embryos display upregulated *SM50* expression at the tips of all the rods, with the exception of the mid-ventral rod, which displays uniform expression patterns. In contrast, MEK-inhibited embryos still display uniform *SM50* expression across the skeletogenic cell population. This suggests that ERK signaling is essential for the localized upregulation of *SM50* at the tips of the rods and for its downregulation in the cells in the back. Overall, these findings indicate that ERK activity is essential for the expression of *ets1/2*, *VEGFR* and *SM50* in the cells near tips of the rods, and for the downregulation of *SM50* at the cells in the back of the rods.

## DISCUSSION
The ERK pathway is a key integrator of extracellular and intracellular information, and the driver of various morphogenetic processes (Lavoie et al., 2020). ERK is a key node in the sea urchin

skeletogenic GRN that had apparently evolved by the co-option of ancestral tubulogenesis program for biomineralization (Ben-Tabou de-Leon, 2022; Morgulis et al., 2019). To explore the interactions of ERK with the skeletogenic GRN, we investigated the timing, regulation and downstream targets of ERK signaling during skeletal elongation in *Paracentrotus lividus* sea urchin embryos. We found that ERK is upregualted in skeletogenic cells at 30 hpf and localizes to the tips of skeletal rods by 36 hpf, where it plays a crucial role in skeletal elongation, particularly of the post-oral rods (Figs 1 and 2). VEGF signaling activates ERK specifically in the skeletogenic cells (Fig. 3), and inhibiting VEGF or ERK results in similar changes in the expression levels of key skeletogenic genes (Fig. 4). Additionally, ERK inhibition affects the spatial expression of both regulatory and biomineralization genes (Figs 5 and 6). The implications of these findings are discussed below in the context of skeletal growth in sea urchin

embryos and the evolution of biomineralization in these organisms (see summary in Fig. 7).

Our study is the first to directly link VEGF signaling with ERK activation, driving the elongation of the post-oral and antero-lateral rods (Fig. 7B). This is similar to the role of ERK in driving angiogenetic sprouting towards a VEGF source during vertebrate angiogenesis (Fig. 7D) (Chen et al., 2017; Srinivasan et al., 2009). In sea urchins, localized ectodermal VEGF signaling drives cell migration required for the formation of the post-oral and distal sections of the antero-lateral rods (Duloquin et al., 2007; Adomako-Ankomah and Ettensohn, 2013; Tarsis et al., 2022). VEGFR and MEK inhibitions result in a similar reduction in ERK activity in these two skeletal rods (Fig. 3M), suggesting that VEGF is the primary activator of ERK in these rods. This also suggests that VEGF is signaling through ERK to promote the cell migration that is crucial for the formation of the rods. Taken together, these findings suggest that VEGF-activated ERK may regulate cell migration during sea urchin skeletogenesis, similarly to its role in vertebrate angiogenesis (Chen et al., 2017; Srinivasan et al., 2009), supporting a conservation of these regulatory functions across these systems.

Our results show that VEGF and ERK signaling have a pronounced effect on the expression level of the regulatory genes *ets1/2*, *VEGFR* and *pitx*, and a weaker effect on the expression of the biomineralization genes *SM50* and *SM30* (Fig. 4). The quantitative data are supported by our WMISH data, which shows that ERK inhibition nearly abolishes *ets1/2* and *VEGFR* expression at the cells near the tips of the rods, while only affecting the spatial expression of *SM50* and hardly changing the broad expression of SM30 (Figs 5 and 6). The fast degradation of *VEGFR* mRNA between 28 hpf and 32 hpf and *ets1/2* mRNA between 32 hpf and 36 hpf under MEK

inhibition might be explained by the role of ERK in maintaining mRNA stability (Essafi-Benkhadir et al., 2010; Nagashima et al., 2015). Overall, our findings demonstrate the role of the ERK pathway in controlling the expression of key skeletogenic regulatory genes at the active growth zones, i.e. the tips of the rods.

Although ERK regulates tip-specific gene expression in both sea urchin skeletogenesis and vertebrate angiogenesis, the mechanisms underlying this common outcome are unique to each system (Fig. 7, yellow shading). As mentioned above, vertebrate ERK controls angiogenic tip-stalk differentiation through the Delta-Notch pathway (Bridges and Harris, 2016; Gerhardt et al., 2003; Shin et al., 2016). However, genetic and pharmacological perturbations of the Delta-Notch pathway during sea urchin development affect pigment and neural cell specification, but do not affect skeletal formation and growth (Materna and Davidson, 2012; Sherwood and McClay, 1997, 1999; Slota and McClay, 2018). Early in development, ERK activates a transient *delta* expression in the skeletogenic cells (Chessel et al., 2023), which is important for the specification of the neighboring non-skeletogenic mesoderm into pigment cells (Croce and McClay, 2010; Materna and Davidson, 2012; Ransick and Davidson, 2006; Sherwood and McClay, 1997, 1999). During skeletal elongation, Delta is expressed in the neuroectoderm, and Delta perturbations affect neural specification (Materna and Davidson, 2012; Sherwood and McClay, 1997, 1999; Slota and McClay, 2018). These non-skeletogenic roles of Delta and Notch, combined with the absence of cell-junctions in syncytial network of the skeletogenic cells, negates the use of ERK-mediated Delta-Notch signaling in sea urchin skeletogenesis.

Moreover, we identified a VEGFR-ERK feedback loop that may functionally replace the Delta-Notch signaling pathway in

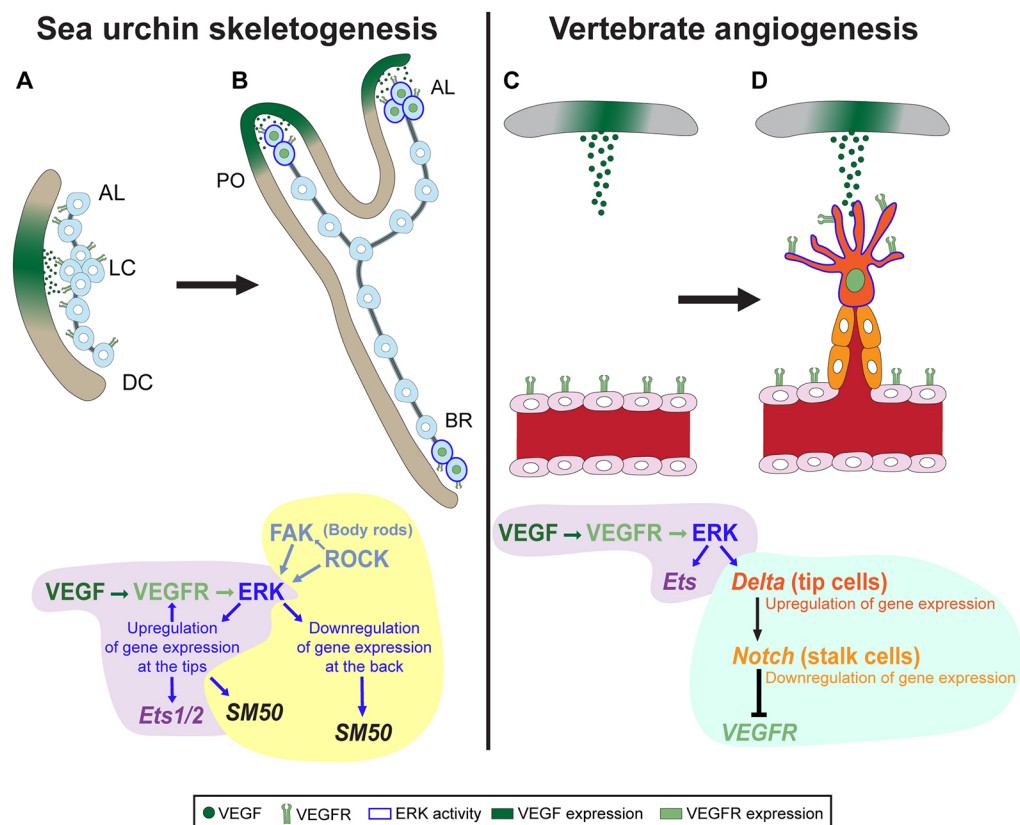

Fig. 7. Comparison between sea urchin skeletogenesis and vertebrate angiogenesis. (A) During sea urchin skeletogenesis, VEGF is expressed in patches of ectodermal cells near lateral cell (LC) clusters, while VEGFR is expressed uniformly in the skeletogenic cells. (B) During skeletal elongation, the binding of VEGF to VEGFR triggers the ERK pathways, which in turn leads to the upregulation of skeletogenic regulatory and biomineralization-specific genes at the tips of the rods. Additionally, ERK receives inputs from FAK and ROCK during the elongation of the body rods (BR). (C) During vertebrate angiogenesis, hypoxic tissues release the VEGF ligand, which binds to VEGFR expressed on endothelial cell membranes. (D) The binding of VEGF to VEGFR activates the ERK pathway, which in turn activates the transcription factor Ets in addition to Delta, which promotes the differentiation of tip cells. Delta then binds to Notch receptors on adjacent cells, triggering their differentiation into stalk cells. These processes drive the sprouting of new blood vessels. AL, antero-lateral rods; DC, dorsal chain rods ; PO, post-oral rods.

controlling differential gene expression at the skeletogenic tip cells of the post-oral and anterolateral rods. *VEGFR* expression in the skeletogenic cell clusters is independent of ERK activity up to 28 hpf (Fig. 5M). However, at 32 hpf, *VEGFR* is nearly abolished from the skeletogenic cells when ERK activity is blocked (Fig. 5N). *ets1/2* expression is also independent of ERK activity up to 32 hpf, when its enrichment in the skeletogenic cell clusters disappears under ERK inhibition (compare Fig. 5B to F) and at 36 hpf, *ets1/2* expression is almost completely lost in this condition (Fig. 5G). ERK skeletogenic activity becomes localized to the tips of the rods at 36 hpf (Fig. 2I-L,Q) and depends on VEGF signaling near the tips of the post-oral and anterolateral rods at 40 hpf (Fig. 3). This suggests that a positive-feedback loop involving VEGF, VEGFR, ERK and potentially the Ets1/2 transcription factor is established during skeletal elongation in the skeletogenic cells closest to the VEGF-secreting ectodermal cells at the post-oral and anterolateral rods (Fig. 7B). This localized feedback loop in the tip cells likely regulates the expression of biomineralization genes such as SM50 in these cells. Therefore, while ERK similarly controls tip-specific differential gene expression in sea urchin skeletogenesis and vertebrate angiogenesis, the regulatory mechanisms that it activates differ between the two systems.

ERK activity at the tips of the body rods is probably regulated by other regulatory inputs in addition to VEGF signaling. The body rods elongate within the dorsal chain of the pre-formed ring of skeletogenic cells and do not receive VEGF signals at their tips (Guss and Ettensohn, 1997). Despite this, ERK activity is still detected at the tips of the body rods (Fig. 2), affecting gene expression there (Figs 5 and 6), with VEGFR inhibition affecting this activity to a lesser extent than MEK inhibition (Fig. 3M). A recent study has shown that FAK and ROCK inhibitions reduce ERK activity at the tips of all skeletal rods, including the body rods (Layous et al., 2025). F-actin, active FAK and Vinculin are found around the spicule rods and are especially enriched at the tips, suggesting the formation of focal adhesion complexes around the spicules (Layous et al., 2025). Possibly, the ERK pathway is capable of integrating both VEGF signaling from the ectoderm, as well as mechanobiology signals emanating from the inner spicule membrane (Fig. 7B).

Overall, our findings provide new insights into the molecular mechanisms governing sea urchin skeletogenesis and enhance our understanding of the evolutionary origins of biomineralization in these organisms. Similar to vertebrate angiogenesis, ERK signaling in sea urchin skeletogenesis is regulated by VEGF and drives differential gene expression at the tips of the elongating rods, including *VEGFR* and *ets*. However, the regulatory mechanisms supporting tip cell gene expression differ between the syncytial skeletogenic lineage and the Delta-Notch driven endothelial tip-stalk differentiation. Furthermore, during sea urchin biomineralization, ERK integrates mechanical inputs and drives additional targets, including biomineralization-specific genes. These changes in ERK signaling might reflect the adaptability of cellular signaling networks as they evolve to support new developmental processes, while maintaining strong connections to ancestrally related molecules.

## MATERIALS AND METHODS
### Animals and embryos
Adult *Paracentrotus lividus* sea urchins were obtained from the Institute of Oceanographic and Limnological Research (IOLR) in Eilat, Israel. To collect gametes, eggs and sperm were harvested by injecting a 0.5 M KCl solution into the adult sea urchins. Embryos were cultured in 0.2 μm-filtered artificial seawater (ASW) at 18°C.

### Imaging
Live embryos and whole-mount *in situ* hybridization experiments were imaged by Zeiss Axioimager 2 (Figs 1, 5 and 6). Fluorescent markers for whole embryos were imaged using a Nikon A1-R Confocal microscope (Figs 2 and 3). Images were edited in Adobe photoshop and Fiji ImageJ. Final figures and illustrations were configured and aligned using Adobe Illustrator.

### Pharmacological inhibitors treatments
ERK (U0126, 9903 Cell Signaling) inhibitor treatment was applied as described previously (Layous et al., 2025) and VEGFR (Axitinib, AG013736, Selleckchem) inhibitor treatment was applied as described previously (Tarsis et al., 2022). Control embryos were cultured in DMSO at the same volume as the inhibitor solution and no more than 0.1% (v/v).

### Immunostaining procedure
Immunostaining for whole embryos were carried out similarly to the method of Winter et al. (2021) with the following antibodies: ERK [Phospho-p44/42 MAPK (ERK1/2) (Thr202/Tyr204) (D13. 14. 4E) XP® Rabbit mAb 4370, Cell Signaling; 1:150] and 6a9 monoclonal antibody (a generous gift from Prof. Charles Ettensohn, Carnegie Mellon University, Pittsburgh, PA, USA, produced in his lab, mouse; 1:5).

### cDNA preparation and QPCR experiments
RNA extraction and cDNA synthesis were performed as described previously (Hijaze et al., 2024). The quantitative polymerase chain reaction (QPCR) procedure was conducted following the methodology described by Tarsis et al. (2022). Changes in gene expression were measured at different conditions at the same time point using ubiquitin as an internal reference gene for normalization (Gildor and Ben-Tabou de-Leon, 2015; Tarsis et al., 2022). A complete list of primer sequences used for the experiments is provided in Table S1.

### Whole-mount *in situ* hybridization procedure
Whole-mount in situ hybridization was conducted in accordance with the methods described by Tarsis et al. (2022). A complete list of primer sequences used for probe generation is provided in Table S2. Probes were added at the following concentrations: 1 ng/μl for ets1/2 and VEGFR probes, 0.7 ng/μl for SM50 probe, and 0.3 ng/μl for SM30 probe.

### Statistical testing
Different statistical tests were run for different experiments, as follows:

1. Non-parametric Mann–Whitney *U*-tests were run to compare the percentage of embryo phenotypes due to small sample sizes ($n$=3). The testing was conducted in IBM SPSS version 27.0.1 and graphs were prepared in Microsoft Excel.

2. Kruskal–Wallis tests (non-parametric one-way ANOVA) along with pairwise tests between groups were used to detect significant changes in the pERK signal under VEGFR or MEK inhibition. Non-parametric testing was used due to the non-normal distribution and/or non-homogeneity of variance. The testing was conducted in IBM SPSS version 27.0.1 and graphs were prepared in Microsoft Excel.

3. One-way ANOVA tests along with Bonefferi post-hoc tests were performed, when possible, to determine the difference in cell number within the skeletogenic rods between VEGFR inhibition, MEK inhibition and control groups. When distribution or variance were not normal, Kruskal–Wallis tests along with pairwise tests between groups were performed. The testing was conducted in IBM SPSS version 27.0.1 and graphs were prepared in Microsoft Excel.

4. One-tailed z-tests were performed to determine significant changes in RNA expression levels in the QPCR experiments. The testing and graph preparation were conducted in Microsoft Excel.

### Acknowledgements
We thank Dr Boris Shklyar from the core microscopy facility at the University of Haifa for technical assistance with confocal microscopy. We also thank Prof. Charles Ettensohn for the gift of the 6a9 antibody and Jenifer Croce for the gift of the SM50 plasmid, and David Ben-Ezra, Michael Kantorovitz and Alvaro Israel for their help with sea urchin handling and maintenance.

**Competing interests**
The authors declare no competing or financial interests.

**Author contributions**
Conceptualization: T.N., M.L., T.G., S.B.-T.d.-L.; Formal analysis: T.N., S.B.-T.d.-L.; Funding acquisition: S.B.-T.d.-L.; Investigation: T.N., S.B.-T.d.-L.; Methodology: T.N., T.G., M.L.; Project administration: T.G.; Supervision: T.G., S.B.-T.d.-L.; Validation: T.N.; Visualization: T.N.; Writing – original draft: T.N., S.B.-T.d.-L.; Writing – review & editing: T.N., T.G., S.B.-T.d.-L.

**Funding**
This work was supported by the Israel Science Foundation (211/20 to S.B.-T.d.-L.). Open Access funding provided by the University of Haifa. Deposited in PMC for immediate release.

**Data and resource availability**
All relevant data can be found within the article and its supplementary information.

**Peer review history**
The peer review history is available online at https://journals.biologists.com/dev/lookup/doi/10.1242/dev.204684.reviewer-comments.pdf

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
