## [Peer Review File · Development (Cambridge, England)]

Regulatory feedback between VEGF and ERK pathways controls tip-cell expression during sea urchin skeletogenesis

Tovah Nehrer, Tsvia Gildor, Majed Layous and Smadar Ben-Tabou de-Leon
DOI: 10.1242/dev.204684

Editor: Cassandra Extavour

Review timeline

Original submission:	28 January 2025
Editorial decision:	24 February 2025
First revision received:	11 March 2025
Accepted:	4 June 2025

Original submission

First decision letter

MS ID#: dev.204684

MS Title: Regulatory feedback between VEGF and ERK pathways controls tip-cell expression during sea urchin skeletogenesis

Authors: Tovah Nehrer; Tsvia Gildor; Majed Layous; Smadar Ben-Tabou de-Leon
Article Type: Research Article

Dear Dr Ben-Tabou de-Leon,

I have now received all the referees' reports on the above manuscript, and have reached a decision. The referees' comments are appended below, or you can access them online: please go to:

As you will see, the referees express considerable interest in your work, but have some significant criticisms and recommend a substantial revision of your manuscript before we can consider publication. If you are able to revise the manuscript along the lines suggested, which may involve further experiments, I will be happy receive a revised version of the manuscript. Your revised paper will be re-reviewed by one or more of the original referees, and acceptance of your manuscript will depend on your addressing satisfactorily the reviewers' major concerns. Please also note that Development will normally permit only one round of major revision. If it would be helpful, you are welcome to contact us to discuss your revision in greater detail. Please send us a point-by-point response indicating your plans for addressing the referees' comments, and we will look over this and provide further guidance.

Please attend to all of the reviewers' comments and ensure that you clearly highlight all changes made in the revised manuscript. Please avoid using 'Tracked changes' in Word files as these are lost in PDF conversion. I should be grateful if you would also provide a point-by-point response detailing how you have dealt with the points raised by the reviewers in the 'Response to Reviewers' box. If you do not agree with any of their criticisms or suggestions please explain clearly why this is so.

Reviewer 1

SUMMARY OF THE ADVANCE MADE IN THIS PAPER AND ITS POTENTIAL SIGNIFICANCE TO THE FIELD

The manuscript authored by Nehrer et al., titled "Conserved pathway, divergent outcome: the role of ERK signaling in Sea urchin larval skeletal development", aims to bridge the gap between VEGFR signaling and ERK in the process of larval skeleton elongation. Convincing evidence provided through the use MEK and VEGFR inhibitors, along with monitoring of ERK activation and skeleton differentiation. The authors also monitored changes in target genes by qPCR, and to a lesser extent, by in situ hybridization. The latter provides spatiotemporal support for the qPCR results. The manuscript concludes with a suggested mechanism explaining the difference between skeletogenesis and angiogenesis. Overall, the experiments were carried out carefully, and the conclusions drawn are supported by the presented data.

SUGGESTIONS TO AUTHORS

Main comment:

I have one main comment about the conclusion. In Fig. 7, the basis for the dissociation of Delta/Notch from the regulatory network is not clear. None of the experiments address Delta/Notch perturbations to conclude an alternative mechanism (alternative to angiogenesis). The finding of positive feedback between dpERK and VEGFR does not necessarily suggest a direct connection, nor does it rule out the involvement of other players. In Fig. 4, changes in the Notch transcript levels are checked, but Delta was not monitored. Hence, this alternative mechanism (lack of Delta/Notch involvement) must be either shown experimentally or supported by existing literature.

Minor comments:

Fig. 1- It is difficult to distinguish between the green (antero-lateral) and blue (lateral cells) colored spheres.

Figs. 2,3- The selection of green and black triangles should be changed, especially with the dominance of these colors in the images.

Fig. 4- What are the units on the X-axis?

Figs. 5,6- What other patterns were observed across all treatments that are not represented by the images? For example, in A: the number 42 represents similar images, what about the other 26?

What patterns do they have?

Reviewer 2

This manuscript reports on skeletogenesis of the sea urchin larva with a focus on the RTK pathway involving VEGF and ERK a downstream transduction molecule that must be phosphorylated when the VEGF is active. They show the pattern of ERK activity with an activated ERK antibody. An attempt is made to argue that of tubulogenesis, as a process is co-opted in production of the skeleton. Yes, the skeletogenic syncytium forms a sort of tube within which the skeleton grows, and yes, the RTK pathway is used in other systems where a tube forms, but the RTK pathway is used for a myriad of processes. They report finding exceptions to the tubulogenesis idea and skeletogenesis and make the claim that modifications are expected in evolution. You can't argue that last point, but this argument is pointless because they can't prove it, and who is to know - tubulogenesis and skeletogenesis, and numerous other structures owe some of their regulation to the RTK signal transduction pathway. So, co-option is questionable here in terms of adopting a process used by one organism for a purpose that is different than that ancient organism. Since the RTK pathway is so prevalent, the original use of that pathway is unknown, and the thousands of uses of the pathway all are co-opted from that original use, so many adaptations that it is pointless to call it co-option. And it is pointless to try and connect skeletogenesis to a tube-making function. Yes, there is a tube, but the data here have nothing to do with formation of that tube. They focus on formation of a biomineral. If I ask what forms the tube? The answer to that isn't given by the data in this paper. Consequently, I strongly recommend that the authors rewrite the paper and eliminate that aspect.

Abstract: It is interesting to think of possible evolutionary processes. However, to state bluntly that an ancient program has been co-opted by vertebrates is overstepping an understanding of evolutionary processes. To this reviewer it is OK to suggest that an ancient program has been co-opted, or indicate that it has been proposed that such a co-option occurred, but the burden of proof must be high for the certainty indicated in this abstract. Also, it is easy to interpret your statement that the vertebrate adaptation is what was co-opted by sea urchins. I know that isn't what you mean but the language regarding who adapted what from whom is a bit confusing. Further, you state that ERK picked up regulation from a biomineralization pathway over and above

its role in an angiogenesis pathway. Again, while that may have occurred, you don't know that is true. You can suggest that possibility, but again it is only a possibility.

Line 66: tri-radiate?

Line 68: First, it is an idea - unproven - that the cavity generated by the skeletal cell syncytium is related to tubulogenesis used for vascularization. There are a huge number of differences between the two so you will find many people who disagree with your idea. That doesn't eliminate your idea, but since it is just an idea you are able to state it as such, but care should be taken to indicate it is an idea. That means your sentence beginning with "Therefore...." will be met with disbelief by most readers. Since the tubular cavity formed by the skeletogenic syncytia is so different from the details of the angiogenesis process you must reconsider the theme you propose, or you will lose your audience. Sure, both use VEGF, ERK, and its transcription factors, but you understand all three molecules are used for many mechanisms in development and physiology. I recommend your read Lavoie, et al., (2020) Nature Reviews of Molecular Cell Biology 21:607-632. It covers Receptor tyrosine kinases, ERK, and its factors playing a role in numerous processes in both normal and disease states.

Further, to this reviewer, it would be of value to state your reasons for thinking along these lines is based on several observations previously noted in your earlier publication. I note however, that all of your data is at sites of skeletal growth, and none of your observations test anything to do with formation of the tube in which the skeleton grows. That to me, says "why go there?" You produce data that has not previously been known in the sea urchin system - that should be enough to get a publication. Why don't you simply indicate that the RTK-MAPK-its family of transcription factors are used for many processes, including vasculogenesis, (and others), and skeletogenesis. Here, your intention is to clarify details of HOW that circuit is used in biomineralization of the skeleton - going beyond the earlier observations that it is used. It gets around all the controversy and frankly, the disbelief that your co-option thesis generates.

Line 214: The difference between your experiments here and previous experiments is that the two inhibitors were used under the same conditions, in the same batch of embryos. Also the sites of ERK activity are original. I think the strength of this manuscript is to show where, at any given time, ERK is active, and to also show that the activity correlates with activity of VEGF. You learn where during growth of the rod is ERK activated by VEGF and where is it not.

Line 264: To be consistent I think it would be better to stick with calling it the VEGF pathway instead of switching to VEGFR pathway. Overall, it looks as if knockdown of VEGF signaling affects skeletogenic cells by reducing its1/2, as expected, and therefore reducing production of skeletal matrix proteins as well as a strong reduction of the VEGFR. How does Pitx fit into the skeletogenesis story? I went to your paper on that to find out- you might quickly summarize what you know.

Line 341: I don't think this is correct. Fig S1 shows that under all conditions the total number of skeletogenic cells remains the same - therefore there is no proliferation, and in fact many studies have indicated that the number of skeletogenic cells remains constant with one division occurring after ingress to reach 64 (32 in *S. purpuratus*) until after the larva begins to feed. That means the reduction of cells in the post oral rod is due to failure to migrate there rather than failure to proliferate. Further, as above, since proliferation hasn't been demonstrated - it erases any relationship to vertebrate cell division during angiogenesis.

Line 364: For ERK activity to be upregulated through VEGF signaling you would first need the VEGFR. So, the feedback you suggest may be real but the upregulation of VEGFR mRNA expression that occurs initially cannot happen by VEGF signaling since there is no VEGFR to receive that signal. You might look into citing Lepage who reports ERK activation in the absence of VEGF signaling. In any case, by now the connection you try to make with vasculogenesis is so hollow it takes away from the data you do have.

Line 382. Mechanosensing? Just because focal adhesion is thought to be used because FAK is expressed doesn't mean that you can say this sentence. Yes, another signal is probably used at the Scheitel but to conclude that mechanosensing is involved is not acceptable. In fact, if that is the case, why doesn't the body rod tips use mechanosensing since the tension pushing out the

epithelium there leads to more punctures of the skeleton through the epithelium than does the Scheitel.

First revision

Author response to reviewers' comments

Dear reviewers,

Thank you very much for thoughtfully reading our manuscript and providing us with critical feedback. We did our best to address all your comments and believe that the current version of the paper is clearer and more scientifically sound and thank you for that. Below are our point-by-point answers to your comments and suggestions.

Reviewer 1: SUGGESTIONS TO AUTHORS

Main comment:

I have one main comment about the conclusion. In Fig. 7, the basis for the dissociation of Delta/Notch from the regulatory network is not clear. None of the experiments address Delta/Notch perturbations to conclude an alternative mechanism (alternative to angiogenesis). The finding of positive feedback between dpERK and VEGFR does not necessarily suggest a direct connection, nor does it rule out the involvement of other players. In Fig. 4, changes in the Notch transcript levels are checked, but Delta was not monitored. Hence, this alternative mechanism (lack of Delta/Notch involvement) must be either shown experimentally or supported by existing literature.

- Sea urchin Delta-Notch signaling was studied in previous works that show that this pathway is regulating the specification on non-skeletogenic cells, early in development, and neural specification later. Importantly, when Delta-Notch is perturbed genetically or pharmacologically, skeletal growth is completely normal. In the previous version of the paper we described these findings in the introduction, but their proper place is in the discussion, when we present our model. We moved this paragraph into the discussion and added more references to previous works that clarify this point better (lines 323-339).

Minor comments:

Fig. 1- It is difficult to distinguish between the green (antero-lateral) and blue (lateral cells) colored spheres.

- We changed the colors in this figure for better visualization.

Figs. 2,3- The selection of green and black triangles should be changed, especially with the dominance of these colors in the images.

- We marked the outline of these triangles in yellow and white so now it is much easier to distinguish them.

Fig. 4- What are the units on the X-axis?

- The X-axis in Fig. 4 is the ratio between gene mRNA level in the treatment compared to the control, therefore it is a pure number. The caption mistakenly stated that these are MEK inhibited vs. VEGFR inhibited levels, which is wrong. Both are presented compared to control embryos (DMSO treated, lines 491-492)

Figs. 5,6- What other patterns were observed across all treatments that are not represented by the images? For example, in A: the number 42 represents similar images, what about the other 26? What patterns do they have?

- Other embryos showed mild differences of the expression patterns reported here, either

stronger or weaker staining in certain rods which can be explained by asynchrony of the cultures. We added this sentence to the captions of the two figures and include the complete data set with all expression patterns at all time points in all conditions, in supplementary figures 3 and 4.

Reviewer 2: SUGGESTIONS TO AUTHORS

This manuscript reports on skeletogenesis of the sea urchin larva with a focus on the RTK pathway involving VEGF and ERK a downstream transduction molecule that must be phosphorylated when the VEGF is active. They show the pattern of ERK activity with an activated ERK antibody. An attempt is made to argue that tubulogenesis, as a process is co-opted in production of the skeleton. Yes, the skeletogenic syncytium forms a sort of tube within which the skeleton grows, and yes, the RTK pathway is used in other systems where a tube forms, but the RTK pathway is used for a myriad of processes. They report finding exceptions to the tubulogenesis idea and skeletogenesis and make the claim that modifications are expected in evolution. You can't argue that last point, but this argument is pointless because they can't prove it, and who is to know - tubulogenesis and skeletogenesis, and numerous other structures owe some of their regulation to the RTK signal transduction pathway. So, co-option is questionable here in terms of adopting a process used by one organism for a purpose that is different than that ancient organism. Since the RTK pathway is so prevalent, the original use of that pathway is unknown, and the thousands of uses of the pathway all are co-opted from that original use, so many adaptations that it is pointless to call it co-option. And it is pointless to try and connect skeletogenesis to a tube-making function. Yes, there is a tube, but the data here has nothing to do with formation of that tube. They focus on formation of a biomineral. If I ask what forms the tube? The answer to that isn't given by the data in this paper. Consequently, I strongly recommend that the authors rewrite the paper and eliminate that aspect.

- Evolutionary claims are very hard to prove, and we agree with the reviewer that we made a mistake in our presentation, first by presenting the proposed evolutionary scenario as a fact and second, by not providing sufficient arguments to support it. We therefore revised the paper to emphasize that the proposed evolutionary scenario is, indeed proposed, and did our best to explain better what supports this model.
- Briefly: Biomineralization is believed to have evolved independently in different phyla, by the co-option of a phylum specific organic scaffold for biomineralization (Murdock and Donoghue, 2011; Murdock, 2020). Echinoderm calcite skeletons are generated within a syncytial network of cells that form calcite structures within them, and in the echinoderm species that have larval skeletons, the spicules have a tubular structure. The spicules grow by elongating this tubular structure through deposition of mineral ions and matrix protein into the inner membrane of the tube, engulfing the biomineral. Thus, in the sea urchin larva, skeletal growth involved tubulogenesis and biomineralization that practically occur together. To illustrate this process **we added a new figure that shows the elongation of the post-oral rods at 48hpf, Fig. S1.**
- The gene regulatory networks that drive echinoderm skeletogenesis were studied in multiple echinoderm species in the larval stage (sea urchin and brittle star) and in adults (sea star, sea urchin and brittle star adults). These studies demonstrate a common core of regulatory genes that include Ets, VEGFR, Erg and Hex that are also expressed in vertebrate endothelial cells and regulate vascularization. (summarized in this review: (Ben-Tabou de-Leon, 2022)). Less is known about the GRNs that drive vascularization outside vertebrates, but VEGFR and Ets were shown to have a role in vascularization in chordate and protostome species and VEGFR is expressed in tubular organs in Cnidarians (summarized in the last paragraph of the introduction here (Morgulis et al., 2019)). The role of VEGF and Ets in vascularization in bilaterians and the role of VEGF in tubulogenesis in Cnidarians, led us to propose that biomineralization in echinoderms evolved through the co-option of a tubulogenesis GRN. While we didn't include all these details in the paper trying not to overload the reader, we refer to the abovementioned papers and do our best to explain our suggestions better. Below I provide more details in response to your specific comments.

Abstract: It is interesting to think of possible evolutionary processes. However, to state bluntly

that an ancient program has been co-opted by vertebrates is overstepping an understanding of evolutionary processes. To this reviewer it is OK to suggest that an ancient program has been co-opted, or indicate that it has been proposed that such a co-option occurred, but the burden of proof must be high for the certainty indicated in this abstract. Also, it is easy to interpret your statement that the vertebrate adaptation is what was co-opted by sea urchins. I know that isn't what you mean but the language regarding who adapted what from whom is a bit confusing. Further, you state that ERK picked up regulation from a biomineralization pathway over and above its role in an angiogenesis pathway. Again, while that may have occurred, you don't know that is true. You can suggest that possibility, but again it is only a possibility.

- We agree with the reviewer that we were too blunt and not careful enough. We revised the title and abstract significantly, emphasizing the speculative nature of our statement, providing more details on the findings of the paper and eliminating the part about the bone and picking mechanical regulatory cues, as it is not well fitted to the abstract. We also shortened the first introduction paragraph about co-option to one sentence and connected it immediately with biomineralization (lines 28-42).

Line 66: tri-radiate?

- Yes, corrected.

Line 68: First, it is an idea - unproven - that the cavity generated by the skeletal cell syncytium is related to tubulogenesis used for vascularization. There are a huge number of differences between the two so you will find many people who disagree with your idea. That doesn't eliminate your idea, but since it is just an idea you are able to state it as such, but care should be taken to indicate it is an idea. That means your sentence beginning with "Therefore...." will be met with disbelief by most readers. Since the tubular cavity formed by the skeletogenic syncytia is so different from the details of the angiogenesis process you must reconsider the theme you propose, or you will lose your audience. Sure, both use VEGF, ERK, and ets transcription factors, but you understand all three molecules are used for many mechanisms in development and physiology. I recommend your read Lavoie, et al., (2020) Nature Reviews of Molecular Cell Biology 21:607-632. It covers Receptor tyrosine kinases, ERK, and ets factors playing a role in numerous processes in both normal and disease states.

- We agree and therefore erased the sentence beginning with "therefore..." in the previous version. Our claim for co-option is not due to the tubular structure of the spicules, but due to the similarity in the upstream GRNs, and downstream tubulogenesis genes, as explained above. This paragraph now simply explains the development of the spicules. (lines 43-58). We are familiar with the excellent review on ERK by Lavoie, et al., (2020), and should have cited it. We added the citation to the current version.

Further, to this reviewer, it would be of value to state your reasons for thinking along these lines is based on several observations previously noted in your earlier publication. I note however, that all of your data is at sites of skeletal growth, and none of your observations test anything to do with formation of the tube in which the skeleton grows. That to me, says "why go there?" You produce data that has not previously been known in the sea urchin system - that should be enough to get a publication. Why don't you simply indicate that the RTK-MAPK-ets family of transcription factors are used for many processes, including vasculogenesis, (and others), and skeletogenesis. Here, your intention is to clarify details of HOW that circuit is used in biomineralization of the skeleton - going beyond the earlier observations that it is used. It gets around all the controversy and frankly, the disbelief that your co-option thesis generates.

- I hope that what I wrote above explains that it is not just any RTK-MAPK-Ets interactions, but VEGF-MAPK-Ets + Hex, Erg, Tel, FoxO and other members of this GRN that are shared between vertebrate vascularization and sea urchin skeletogenesis + VEGF participation in tubulogenesis outside of deuterostomes. Also - since the spicule is a tube filled with mineral, in the sea urchin skeletal growth involves tubulogenesis and biomineralization together (See Fig. S1).
- I added explanations about our previous studies and toned down all our descriptions of this

theory throughout the introduction and in the rest of the paper. However, I would like to keep the co-option hypothesis, as it adds an important evolutionary dimension and context to our developmental story, and I believe we have enough support to allow us to keep it as an hypothesis.

Line 214: The difference between your experiments here and previous experiments is that the two inhibitors were used under the same conditions, in the same batch of embryos. Also the sites of ERK activity are original. I think the strength of this manuscript is to show where, at any given time, ERK is active, and to also show that the activity correlates with activity of VEGF. You learn where during growth of the rod is ERK activated by VEGF and where is it not.

➤ I agree, thank you!

Line 264: To be consistent I think it would be better to stick with calling it the VEGF pathway instead of switching to VEGFR pathway. Overall, it looks as if knockdown of VEGF signaling affects skeletogenic cells by reducing *ets1/2*, as expected, and therefore reducing production of skeletal matrix proteins as well as a strong reduction of the VEGFR. How does *Pitx* fit into the skeletogenesis story? I went to your paper on that to find out- you might quickly summarize what you know.

➤ We removed the extra R to make it VEGF pathway (line 233).

Line 341: I don't think this is correct. Fig S1 shows that under all conditions the total number of skeletogenic cells remains the same - therefore there is no proliferation, and in fact many studies have indicated that the number of skeletogenic cells remains constant with one division occurring after ingress to reach 64 (32 in *S. purpuratus*) until after the larva begins to feed. That means the reduction of cells in the post oral rod is due to failure to migrate there rather than failure to proliferate. Further, as above, since proliferation hasn't been demonstrated - it erases any relationship to vertebrate cell division during angiogenesis.

➤ I agree with your criticism regarding proliferation and we removed it from the text and left only the description of VEGF and ERK role in cell migration during the growth of the anterolateral and post-oral rods (Lines 308-310).

Line 364: For ERK activity to be upregulated through VEGF signaling you would first need the VEGFR. So, the feedback you suggest may be real but the upregulation of VEGFR mRNA expression that occurs initially cannot happen by VEGF signaling since there is no VEGFR to receive that signal. You might look into citing Lepage who reports ERK activation in the absence of VEGF signaling. In any case, by now the connection you try to make with vasculogenesis is so hollow it takes away from the data you do have.

➤ The exact time and place of the establishment of the feedback loop was not explained well in the previous version and we corrected it in the current version, lines 340-356. Indeed, VEGFR expression in the skeletogenic cells does not depend on ERK signaling at 28hpf (Fig. 5I, M), and becomes dependent only at 32hpf (Fig. 5J, N). We showed a dependence of ERK activity in VEGF signaling at 40hpf, thus, the VEGF-ERK feedback loop kicks in only after 30hpf, when the anterolateral and post-rods begin to elongate.

Line 382. Mechanosensing? Just because focal adhesion is thought to be used because FAK is expressed doesn't mean that you can say this sentence. Yes, another signal is probably used at the Scheitel but to conclude that mechanosensing is involved is not acceptable. In fact, if that is the case, why doesn't the body rod tips use mechanosensing since the tension pushing out the epithelium there leads to more punctures of the skeleton through the epithelium than does the Scheitel.

➤ You are correct that the formation of focal adhesion around the spicule does not, on its own, indicate mechanosensing, and I feel that we explained this poorly here and in the introduction. We therefore revised the section about the FAK-ROCK-ERK circuit in the introduction (lines 111-120) and in the discussion (lines 357-368). Our recent paper had shown that in isolated skeletogenic cells, culturing on soft substrate reduces skeletal

growth rate and increases spicule branching, suggesting that this process is mechanosensitive. We further showed that when the spicule grain is formed it is coated by active FAK and F-actin, suggesting that the cells recognize the spicule, possibly due to its high rigidity. FAK, F-actin and Vinculin are enriched at the tips of the growing spicule (the active growth zone), FAK activity depends on ROCK activity, and inhibition of FAK or ROCK decreases ERK activity at the tips of all rods including the body (Layous et al, PNAS 2025). Interestingly, similar interactions between vertebrate FAK, ROCK, and ERK drive the differentiation of vertebrate osteoblast from mesenchymal stem cells, on hard substrates (Kanno et al., 2007; Khatiwala et al., 2009; Salaszyk et al., 2007; Shih et al., 2011). This led us to propose that a common mechanotransduction pathway was independently activated during the evolution of biomineralization in echinoderms and vertebrates. In the current version we left the facts of ERK activation by FAK and ROCK, which are relevant to ERK activity in the skeletogenic cells and explain which cues are replacing VEGF signaling in the body rods, and reduced the emphasize of mechanosensing, not to overwhelm the readers.

References

- Ben-Tabou de-Leon, S. 2022. The Evolution of Biomineralization through the Co-Option of Organic Scaffold Forming Networks. *Cells*. 11.
- Kanno, T., T. Takahashi, T. Tsujisawa, W. Ariyoshi, and T. Nishihara. 2007. Mechanical stress-mediated Runx2 activation is dependent on Ras/ERK1/2 MAPK signaling in osteoblasts. *J Cell Biochem*. 101:1266-1277.
- Khatiwala, C.B., P.D. Kim, S.R. Peyton, and A.J. Putnam. 2009. ECM compliance regulates osteogenesis by influencing MAPK signaling downstream of RhoA and ROCK. *J Bone Miner Res*. 24:886-898.
- Morgulis, M., T. Gildor, M. Roopin, N. Sher, A. Malik, M. Lalzar, M. Dines, S. Ben-Tabou de- Leon, L. Khalaily, and S. Ben-Tabou de-Leon. 2019. Possible cooption of a VEGF-driven tubulogenesis program for biomineralization in echinoderms. *Proc Natl Acad Sci U S A*. 116:12353-12362.
- Murdock, D.J., and P.C. Donoghue. 2011. Evolutionary origins of animal skeletal biomineralization. *Cells Tissues Organs*. 194:98-102.
- Murdock, D.J.E. 2020. The 'biomineralization toolkit' and the origin of animal skeletons. *Biol Rev Camb Philos Soc*. 95:1372-1392.
- Salaszyk, R.M., R.F. Klees, W.A. Williams, A. Boskey, and G.E. Plopper. 2007. Focal adhesion kinase signaling pathways regulate the osteogenic differentiation of human mesenchymal stem cells. *Exp Cell Res*. 313:22-37.
- Shih, Y.R., K.F. Tseng, H.Y. Lai, C.H. Lin, and O.K. Lee. 2011. Matrix stiffness regulation of integrin-mediated mechanotransduction during osteogenic differentiation of human mesenchymal stem cells. *J Bone Miner Res*. 26:730-738.

Second decision letter

MS ID#: dev.204684R1

MS Title: Regulatory feedback between VEGF and ERK pathways controls tip-cell expression during sea urchin skeletogenesis

Authors: Tovah Nehrer; Tsvia Gildor; Majed Layous; Smadar Ben-Tabou de-Leon

Article Type: Research Article

Dear Dr Ben-Tabou de-Leon,

I am happy to tell you that your manuscript has been accepted for publication in Development, pending our standard publication integrity checks.

Comments from the Reviewers:

Reviewer 1: SUMMARY OF THE ADVANCE MADE IN THIS PAPER AND ITS POTENTIAL SIGNIFICANCE TO THE FIELD

The authors addressed my concerns in the revised version of the manuscript.

SUGGESTIONS TO AUTHORS

None.

Reviewer 2: SUMMARY OF THE ADVANCE MADE IN THIS PAPER AND ITS POTENTIAL SIGNIFICANCE TO THE FIELD

A crucial aspect of larval skeletogenesis in the sea urchin is to learn how the biomineralization of the calcium carbonate skeleton grows. Here the investigators show a positive feedback mechanism at the skeletal tips that involve VEGF signaling that activates ERK-mediated signal transduction. This activates expression of the VEGFR at the skeletal tip that promotes further signaling to extend the skeleton. ERK-mediated signal transduction further is necessary for expression of Ets1/2, a transcription factor, and SM50, a known skeletal matrix protein. These data therefore are a significant addition to an understanding of how biomineralization occurs in this model system.

SUGGESTIONS TO AUTHORS. In reviewing this revision I'm afraid I cannot recommend acceptance. In my first review I strongly recommended a significant reduction in the use of the so-called co-option of VEGF, ERK and Ets from a tubulogenesis origin. Instead, they seem to have doubled down on the idea that the hollow space inside the skeletogenic syncytia originates from the same ancestral mechanism that forms a tube during vascularization. I indicated that numerous mechanisms use these three molecules and to single out vascularization as the central theme of the paper sends the wrong message. They have data that can form the backbone of a good paper without ever mentioning the tubulogenesis possibility and they should go in that direction.

In sea urchin skeletogenesis the hollow in which the biomineralization occurs is within a syncytium. In vasculogenesis there is no syncytium involved. In sea urchin skeletogenesis, the tube that forms is produced by a cell shape change so that each cell in the syncytium extends a fold that curves around to form the tube. That process is completely different from vasculogenesis in which multiple cells form a space in between the cells, and that becomes the lining of the endothelium. In skeletogenesis, spicule matrix proteins are exocytosed as the biomineral is deposited. In vasculogenesis, the exocytosis that occurs lays down a basal lamina that becomes the lining of the endothelium. So, the mechanism of forming a tube is completely different in the two processes. Further, they show that the three molecules are used as part of the production of the biomineral, not in the production of the tube.

Yet, the authors seem stuck on the notion that because VEGF, Ets, and ERK are used in both systems that they must be somehow related. As I indicated in the first review, there are many morphogenetic processes that use these three proteins as part of an RTK signal transduction system. I am adamant here not because the experimental data are flawed, but I want to protect the authors from a deep disbelief that will be exercised by any reader of this paper. I strongly recommend that they reduce this idea of the use of these three molecules for both processes to a paragraph in the discussion. As an example of the distraction of the tubulogenesis idea, one needs only to start with the abstract. Before going into any data, they lead off with 5 lines reflecting this shared tubulogenesis idea as if it is the major theme of the paper. Then, when they arrive at the "Here we reveal...." the story switches to what they do experimentally in the research. To me that could be built into a fine story. As above, it is a story that adds to our understanding of biomineralization. Yet, rather than build the theme of the paper around that, the abstract goes back to this idea of co-option to bring it to a conclusion. In other words, the theme they try to stress is an evolutionary relationship between vasculogenesis and biomineralization and that is to this reviewer a direction that is flawed.